# Mechanistic model for human brain metabolism and its connection to the neurovascular coupling

**Nicolas Sundqvist**[1], **Sebastian Sten**[2,3¤], **Peter Thompson**[1], **Benjamin Jan Andersson**[1], **Maria Engström**[2,3], **Gunnar Cedersund**[1,3]*

**1** Department of Biomedical Engineering, Linköping University, Linköping, Sweden, **2** Department of Health, Medicine and Caring Sciences, Linköping University, Linköping, Sweden, **3** Center for Medical Image Science and Visualization (CMIV), Linköping University, Linköping, Sweden

¤ Current address: Drug Metabolism and Pharmacokinetics, Research and Early Development, Cardiovascular, Renal and Metabolism (CVRM), BioPharmaceuticals R&D, AstraZeneca, Gothenburg, Sweden

* gunnar.cedersund@liu.se

**Data Availability Statement:** All supplementary material required for reproducing the results is

## Abstract

The neurovascular and neurometabolic couplings (NVC and NMC) connect cerebral activity, blood flow, and metabolism. This interconnection is used in for instance functional imaging, which analyses the blood-oxygen-dependent (BOLD) signal. The mechanisms underlying the NVC are complex, which warrants a model-based analysis of data. We have previously developed a mechanistically detailed model for the NVC, and others have proposed detailed models for cerebral metabolism. However, existing metabolic models are still not fully utilizing available magnetic resonance spectroscopy (MRS) data and are not connected to detailed models for NVC. Therefore, we herein present a new model that integrates mechanistic modelling of both MRS and BOLD data. The metabolic model covers central metabolism, using a minimal set of interactions, and can describe time-series data for glucose, lactate, aspartate, and glutamate, measured after visual stimuli. Statistical tests confirm that the model can describe both estimation data and predict independent validation data, not used for model training. The interconnected NVC model can simultaneously describe BOLD data and can be used to predict expected metabolic responses in experiments where metabolism has not been measured. This model is a step towards a useful and mechanistically detailed model for cerebral blood flow and metabolism, with potential applications in both basic research and clinical applications.

## Author summary

The neurovascular and neurometabolic couplings are highly central for several clinical imaging techniques since these frequently use blood oxygenation (the BOLD signal) as a proxy for neuronal activity. This relationship is described by the highly complex neurovascular and neurometabolic couplings, which describe the balancing between increased metabolic demand and blood flow, and which involve several cell types and regulatory

provided in the supplementary material and is also available at: https://gitlab.liu.se/nicsu70/nvc_metabolismmodel.

**Funding:** This work was supported by the Swedish Research Council (2018-05418 and 2018-03319, GC; 2018-03391, ME). Additional support came from CENIIT (15.09, GC), from the Swedish foundation for strategic research (ITM17-0245, GC), from SciLifeLab and KAW (2020.0182, GC), from the H2020 project PRECISE4Q (777107, GC), from the Swedish Fund for Research without Animal Experiments (F2019-0010, GC), from the Swedish Brain Foundation (ME), from ELLIIT (GC), from VisualSweden (GC), and from VINNOVA (2020-04711, GC). The funders have had no role in the study design, data collection and analysis, decision to publish, or preparing the manuscript.

**Competing interests:** The authors have declared that no competing interests exist.

systems, which all change dynamically over time. While there are previous works that describe the neurovascular coupling in detail, neither we nor others have developed connections to corresponding mechanistic models for the third aspect, the metabolic aspect. Furthermore, magnetic resonance spectroscopy (MRS) data for such modelling readily is available. In this paper we present a minimal mechanistic model that can describe the metabolic response to visual stimuli. The model is trained to describe experimental data for the relative change in metabolic concentrations of several metabolites in the visual cortex during stimulation. The model is also validated against independent validation data, that was not used for model training. Finally, we also connect this metabolic model to a detailed mechanistic model of the neurovascular coupling. Showing that the model can describe both the metabolic response and a neurovascular response simultaneously.

# 1 Introduction

The brain is the central organ of the nervous system in humans and it has many complex functions. Despite that it only constitutes approximately 2% of the body weight of an adult individual, the brain is responsible for 20–25% of the body's overall energy consumption [1,2]. The brain's main source of energy comes from oxidation of glucose. Therefore, the brain requires a continuous supply of glucose and oxygen and while the brain can metabolise stored glycogen to produce glucose, part of the required glucose and all of the oxygen is delivered via the blood [3–5]. For these reasons, the cerebral metabolic glucose consumption is directly linked to the regional cerebral blood flow [6,7]. The regional cerebral blood flow is, in turn, tightly coupled to the neuronal activity [8]. An increase in neuronal activity will lead to an increased regional cerebral blood flow, which provides an increased supply of glucose. These couplings between the neuronal, hemodynamic, and metabolic activity are commonly known as the neurovascular coupling (NVC) and the neurometabolic coupling (NMC).

More specifically, the NVC describes how neuronal activity affects cerebral blood volume (CBV) and cerebral blood flow (CBF), while the NMC describes how neuronal activity affects the cerebral metabolic activity such as the cerebral metabolic rate of oxygen ($CMRO_2$) [9–11]. The NVC is a cornerstone of functional magnetic imaging (fMRI), which is a key method for understanding how different stimuli affect regional neuronal activity [12]. Data from fMRI is based on analysis of a blood-oxygenation-level-dependent (BOLD) signal [12,13]. The BOLD signal is governed by the regional balance between the hemodynamic and metabolic activity i.e., by the balance between oxygen supply and oxygen consumption. Both the supply of and the demand for oxygen are regulated by the neuronal activity. This connection between oxygen regulation and neuronal activity is the basis for why the BOLD signal is used as a proxy for the neuronal activity [14,15]. However, using the BOLD signal to draw correct conclusions is not straightforward since the crosstalk between hemodynamic, metabolic, and neuronal activity is nonlinear, time-varying, and partially unknown. One approach to deal with this complexity is mathematical modelling.

Mathematical modelling of the NVC has evolved over time. Initial mathematical modelling of the NVC relied on statistical interpretations of the canonical hemodynamic response function [16–18] (Fig 1A). These models provide estimates of the location and timing of neuronal activity, given a measured BOLD response, but fail to provide any mechanistic insight into the NVC, and the interplay between BOLD, CBV, $CMRO_2$, etc. To remedy this, later mathematical modelling included some mechanistic insight and resulted in the Balloon model, developed by Buxton *et al.* 1998 [19–21]. The Balloon model describes the interplay between neuronal

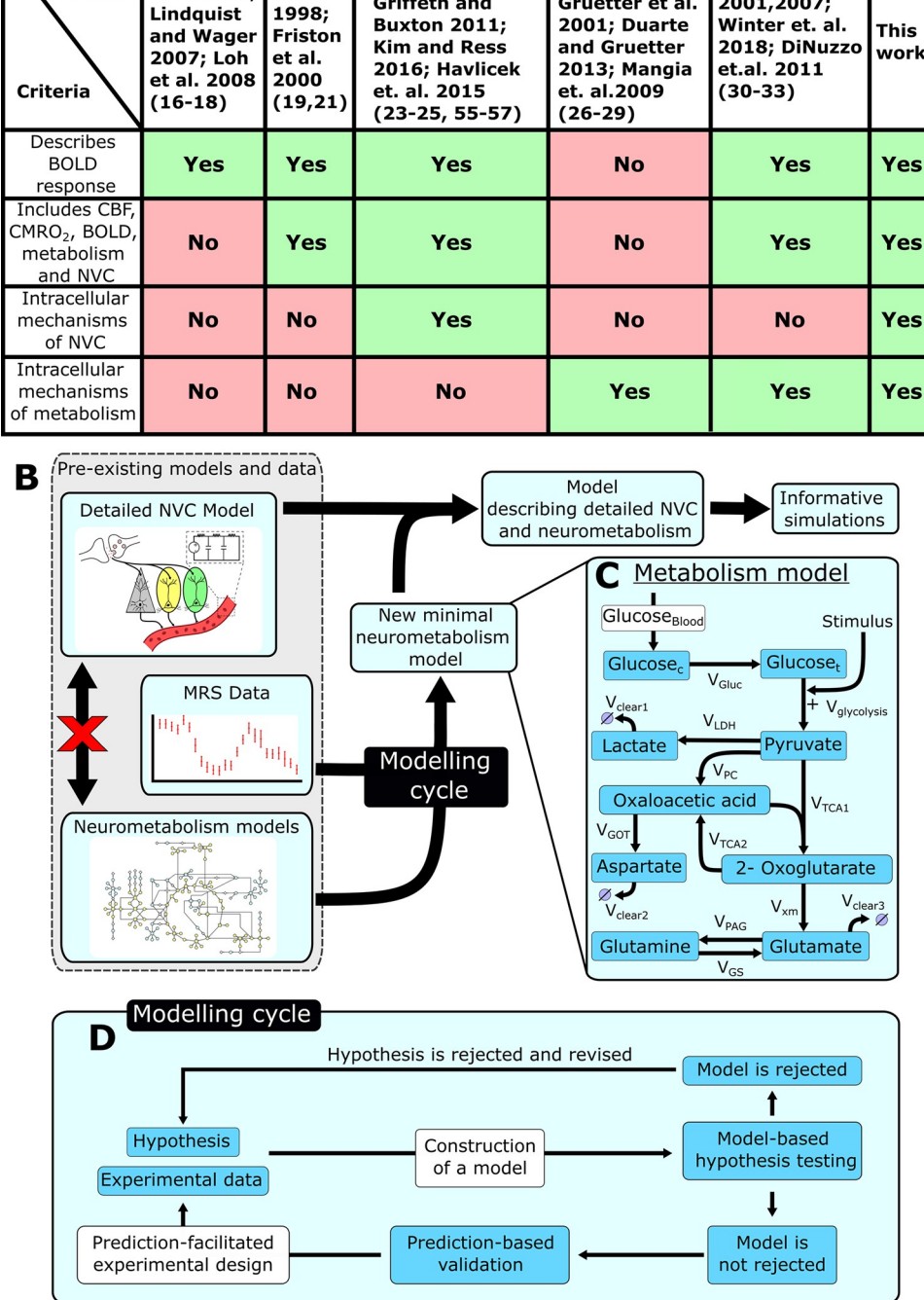

| A. Models / Criteria | Lindquist et al.2009; Lindquist and Wager 2007; Loh et al. 2008 (16-18) | Buxton et al. 1998; Friston et al. 2000 (19,21) | Sten et al. 2017 ,2020,2021; Griffeth and Buxton 2011; Kim and Ress 2016; Havlicek et. al. 2015 (23-25, 55-57) | Aubert and Costalat 2007; Gruetter et al. 2001; Duarte and Gruetter 2013; Mangia et. al.2009 (26-29) | Aubert et. al. 2001,2007; Winter et. al. 2018; DiNuzzo et.al. 2011 (30-33) | This work |
|---|---|---|---|---|---|---|
| Describes BOLD response | Yes | Yes | Yes | No | Yes | Yes |
| Includes CBF, CMRO₂, BOLD, metabolism and NVC | No | Yes | Yes | No | Yes | Yes |
| Intracellular mechanisms of NVC | No | No | Yes | No | No | Yes |
| Intracellular mechanisms of metabolism | No | No | No | Yes | Yes | Yes |

**Fig 1. Overview of the modelling work presented herein.** A. A table summary of different models and what aspects of the neurovascular coupling they cover [16–19,21,23–33,55–57]. B. A schematic overview of how this work connects pre-existing models for the NVC with a description for the cerebral metabolism and how this new interconnected model can be used for informative simulations. C: A detailed illustration of the metabolism model precented in this work. Neuronal activity triggers increased consumption of glucose, which triggers downstream signaling cascades of different metabolites, which can be captured using MRS. D. A schematic illustration of the modelling cycle used to develop a minimal model.

activity, CBF, CMRO$_2$, and the BOLD response, but does so using phenomenological expressions, and thus lack mechanistic interplays between specific substances and intermediaries (Fig 1A). To take NVC modelling one step further, we have therefore created a new type of systems biology modelling approach, which incorporates such mechanistic interplays into the models (Fig 1B, top). This modelling approach first demonstrated that the BOLD response is not controlled by a negative feedback but by a positive feed-forward mechanism [22]. This initial paper was then extended to also explain the negative BOLD response [23]. Both these papers are based exclusively on human data. Lately, our models have also added cell-specific contributions, by also incorporating both rodent optogenetic and primate data [24,25]. All these more mechanistic NVC models still have a quite simple description of the metabolism involved during neuronal activity. However, regarding metabolism modelling without NVC, there are some more detailed models that describe for instance the compartmentalization of the metabolism between different neurons [26–29]. Furthermore, there are works that integrate the metabolic response with NVC, but these works rely on the phenomenological Balloon model [30–33]. It is fairly common that the metabolic parts of these models utilize magnetic resonance spectroscopy (MRS) data [34–36] (Fig 1B, bottom). However, there are still important MRS data that have not been incorporated in any metabolic model (Fig 1B, middle). Also, no such metabolic model has been integrated into a mechanistically detailed NVC model (Fig 1A).

Herein, we utilize both modelled and non-modelled MRS data to construct a new minimal model for NMC (Fig 1C). We use a minimal modelling approach to gain mechanistic insight into the NMC without introducing unneeded complexity. Statistical tests confirm that this model can describe both estimation data (Figs 2 and 3) and independent validation data (Figs 4 and 5), not used for model training. The model is also connected to our previously developed detailed NVC model [23], and the combined model can still describe the previous BOLD-data. The combined model can be used to obtain simulated predictions for the metabolic activity for complex paradigms, involving multiple stimulations, where MRS is currently not available (Fig 6).

## 2 Methods

### 2.1 Model formulation

The model presented herein is formulated using ordinary differential equations (ODEs) and can in general be denoted as

$$\dot{x} = f(x(t), \theta, u(t), t) \tag{1A}$$

$$x(0) = x_0 \tag{1B}$$

$$\hat{y} = g(x(t), \theta, u(t), t) \tag{1C}$$

where $x$ is a vector describing the model states e.g., the concentrations of the different metabolites, signalling intermediates, etc; where $\dot{x}$ describe the derivative of these states with respect to time, denoted $t$; where $\theta$ is a vector of the constant unknown parameters; where $u$ is the model input, here representing a visual stimulus corresponding to the experimental setup; where $x_0$ is the state values corresponding to an initial time point $t = 0$; where $\hat{y}$ are the observed model properties, here the percental difference in metabolite concentrations as well as the BOLD response; and where $f$ and $g$ are smooth nonlinear functions. This general description of an ODE, as well as the general modelling methods in Sections 2.2, 2.4, and 2.6, are described in a similar way as in many previous papers e.g., [23].

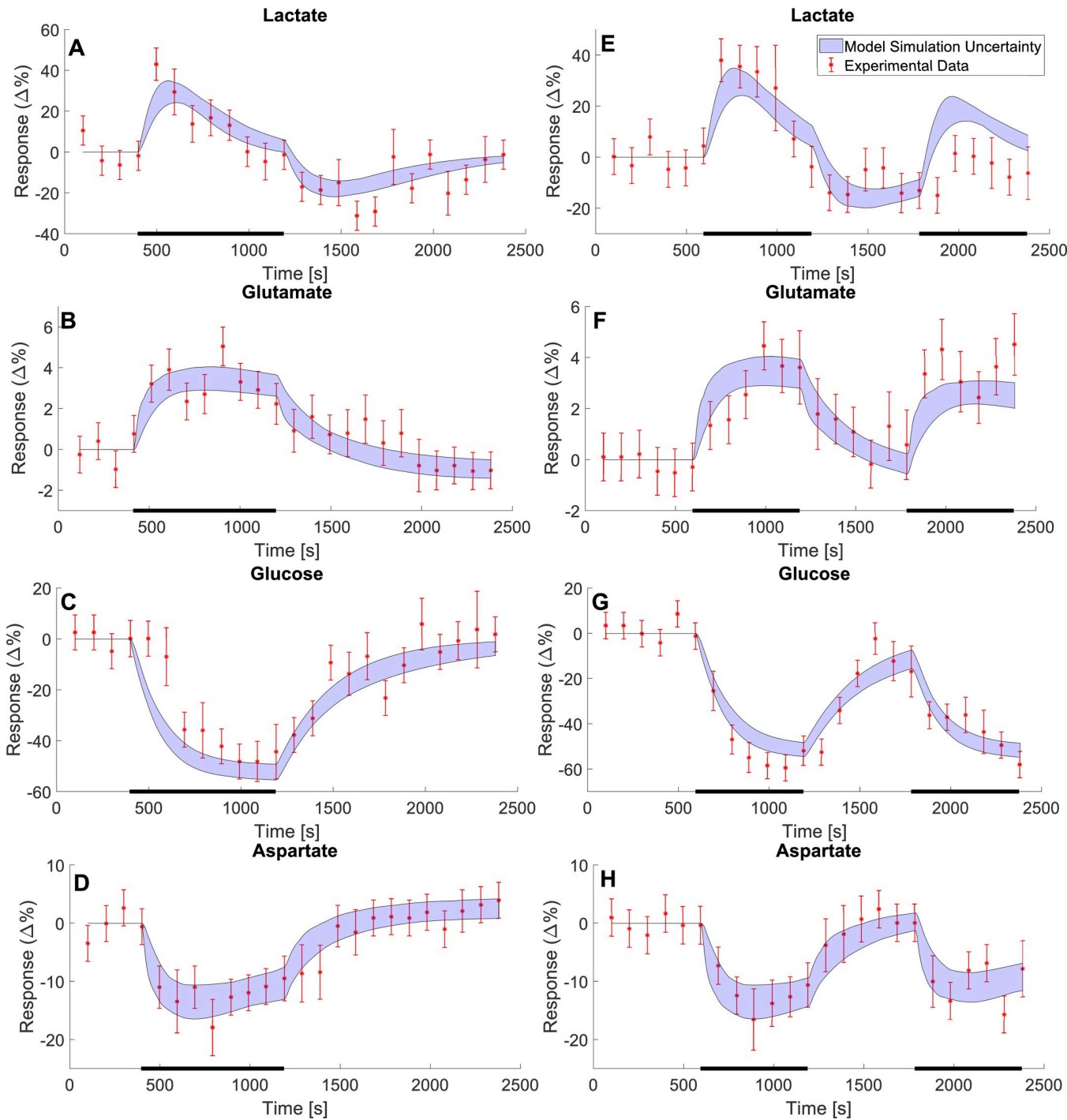

**Fig 2. Model estimation to MRS data showing changes in metabolic concentrations following visual stimuli.** The data gathered from Lin et al. 2012 [34] shows the change in metabolic concentrations as a response to two different visual stimulus paradigms. The first paradigm (A-D) consisted of a single visual stimulus lasting for 13.6 minutes, indicated by the black bar at the bottom of the plots. The second paradigm (E-H) consists of two 9.9-minute periods of stimulus, separated by a 9.9-minute rest period for each. For both stimulation paradigms four metabolites were analysed: lactate (A and E), glutamate (B and F), glucose (C and G), and aspartate (D and H). The data is here illustrated as a relative change from a baseline, with a mean value ± SEM (red error bars). The model estimation is illustrated as the blue shaded areas.

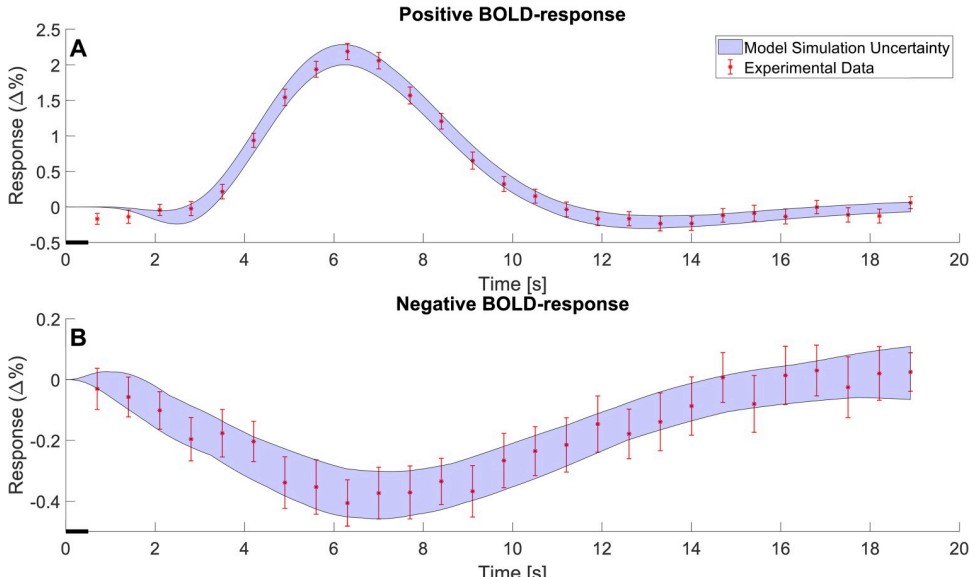

**Fig 3. The model estimation of the BOLD response.** A. The model estimation (blue shaded area) of the positive BOLD response fitted to data (red error bars) to a 0.5 second stimulus. B. The model estimation (blue shaded area) of the negative BOLD response fitted to data (red error bars) to a 0.5 second stimulus. The data is shown as mean values ± SEM and is taken from Sten et al. [23].

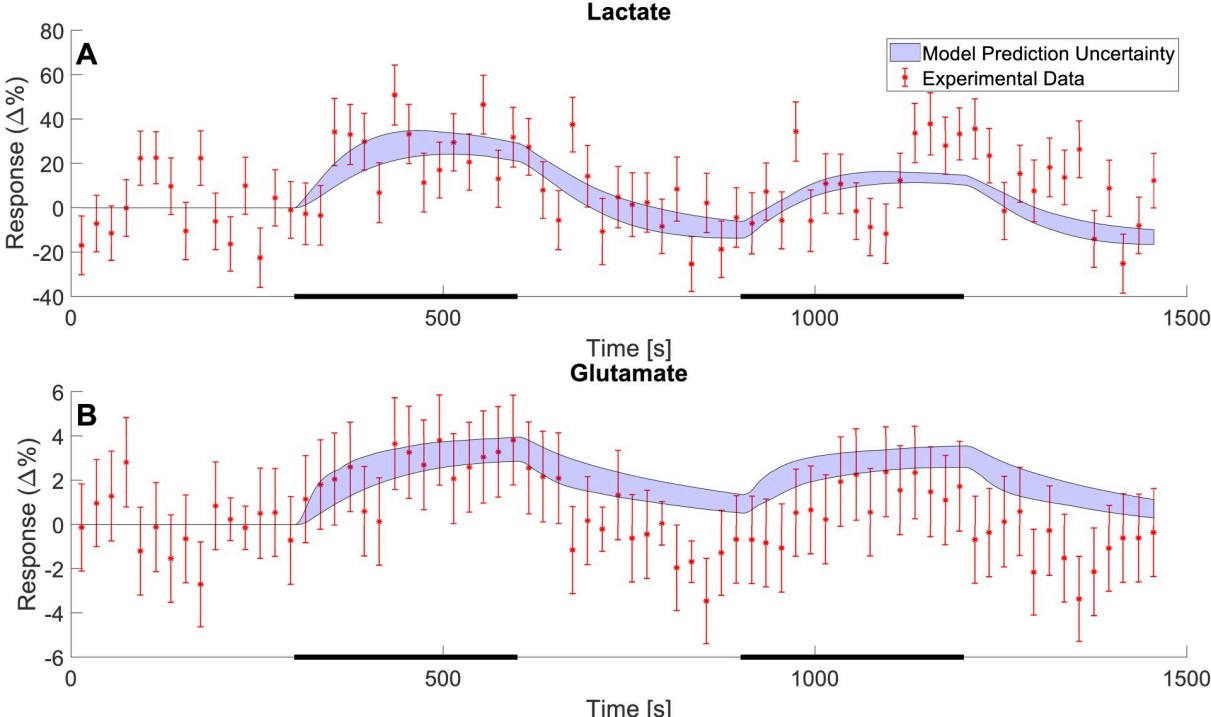

**Fig 4. Model predictions of independent MRS data for lactate and glutamate.** The model predictions (blue shaded areas) of the changes in metabolic concentrations for lactate (A) and glutamate (B) as a response to a double stimulation paradigm. The stimulation consisted of two 5-minute periods of stimulation (black bars) interwoven by a 5-minute rest period. The data is illustrated as mean values ± SEM (red error bars) and was gathered from Schaller et al. 2013 [36].

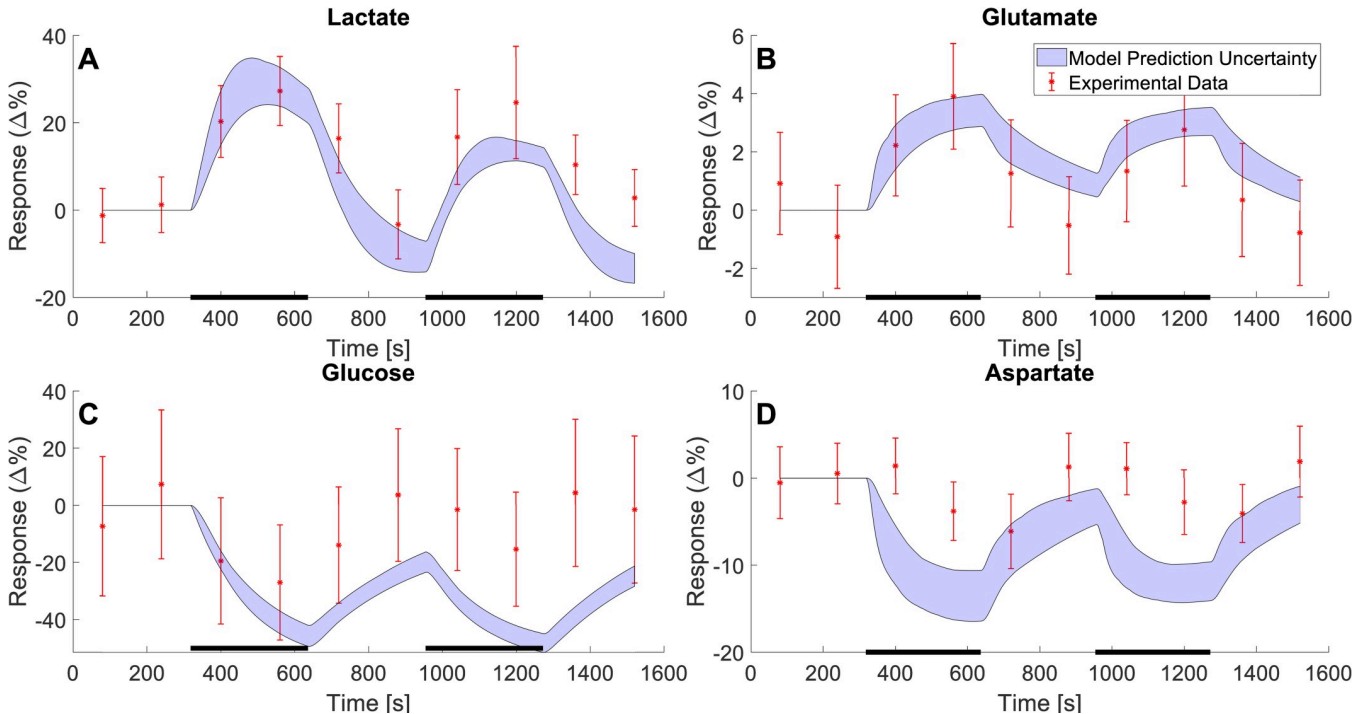

**Fig 5. Model predictions compared with independent MRS data for lactate, glutamate, glucose, and aspartate, not used for training the model.** The model predictions (blue shaded areas) of the changes in metabolic concentrations for lactate (A), Glutamate (B), Glucose (C), and aspartate (D) as a response to a double stimulation paradigm. The stimulation consisted of two 5.3-minute periods of stimulation (black bars) interwoven by a 5.3-minute rest period. The data is illustrated as mean values ± SEM (red error bars) and was gathered from Bednařík et al. 2015 [35].

## 2.2 Development of a minimal model: general workflow

The model presented herein was developed according to an iterative modelling cycle based on hypothesis testing and rejection (Fig 1D). To start with, a falsifiable hypothesis regarding the relevant mechanisms is formulated based on literature and prior knowledge. This hypothesis is then combined with existing experimental data, for the system of interest, to formulate a mathematical model. These mechanistic models are developed as minimal models, which means that they are as simple as possible while still explaining relevant experimental data. Once formulated, each mathematical model is fitted to the experimental data and evaluated with respect to how well the model can describe the data. Such evaluation is done via qualitative inspection of the model simulation but also through quantitative statistical tests; here a $\chi^2$-test. If a model structure cannot satisfactorily describe the data, the model and the corresponding hypothesis is rejected and needs to be revised and then fitted again. If the model structure is not rejected, further model analysis is performed, to determine model identifiability and to generate well-determined predictions, which we call core-predictions [37]. Such core-predictions are ideally generated such that testing them experimentally will generate further biological insights into the performance of the model. Thus, model predictions can act as the basis for further experiment design that in turn generates or find more relevant experimental data. Finally, if this new data agrees with the model predictions, it serves as a support, or partial validation, of the model structure, since such agreement indicates that the model can explain more properties and states than just the experimental data used for model training. A thus validated model can then be used to generate additional predictions, within similar applications as the original development and testing of the model. In this project, we have developed the model in this iterative fashion, and below we present the final accepted model.

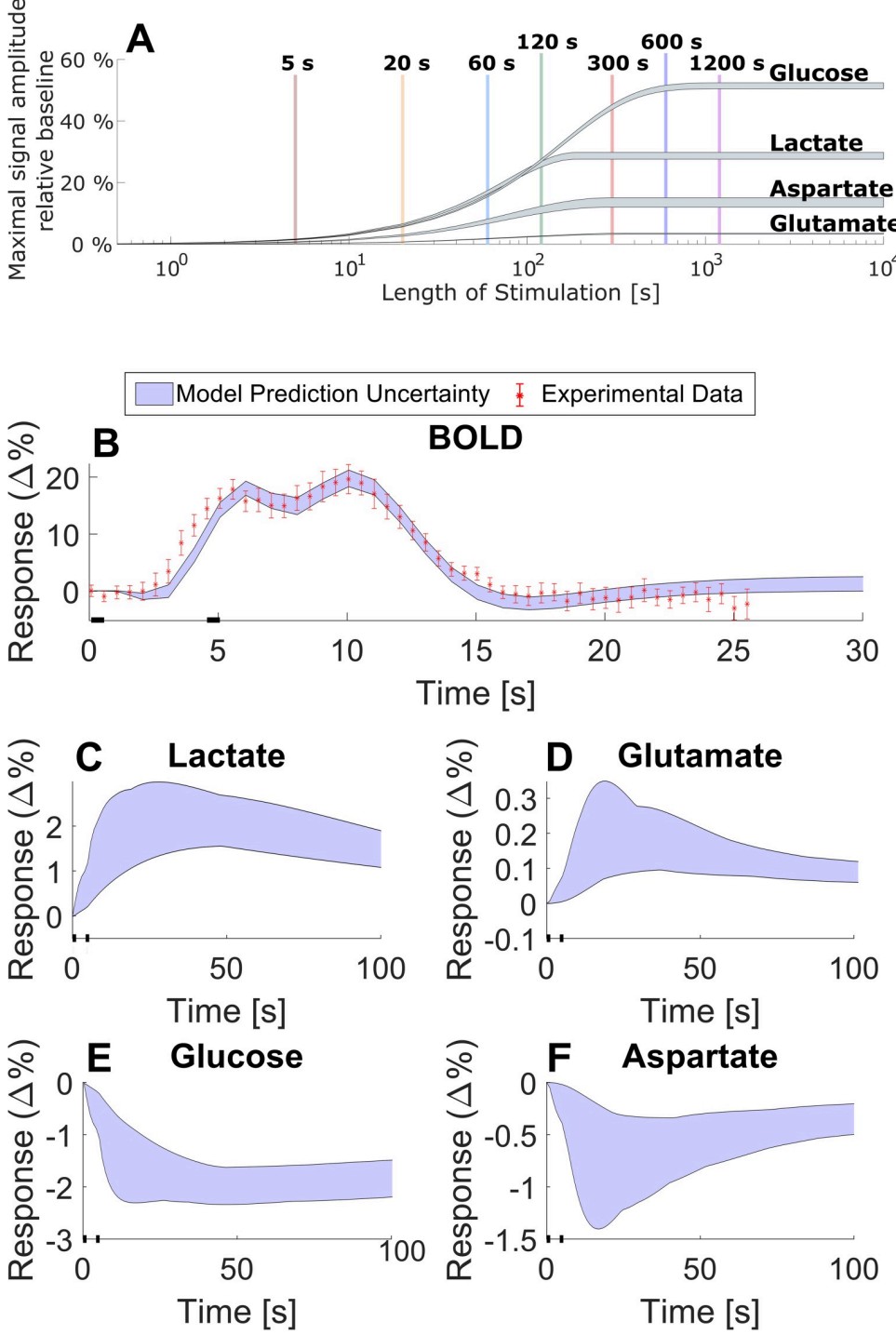

**Fig 6. Examples of the type of simulations that are possible to do with the model.** A. An illustration of how the maximum amplitude of the metabolic response (grey shaded areas) changes with the length of stimulation. B. Shows a model prediction (blue shaded area) of a BOLD response for two short 0.5 seconds stimuli 4 seconds apart and corresponding data (red error bars). The data is gathered from Lundengård et al. [22]. C-F. shows the model predicted metabolic response to the same short double stimuli. C. shows the metabolic response for lactate; D. for glutamate; E. for glucose; and F. for aspartate.

### 2.3 Model structure

**2.3.1 Model structure for the metabolism model.** All events in the model starts with a stimulation. This stimulation represents a visual stimulus and is implemented in the form of a simple step function [23,24,38,39] formulated as:

$$u(t) = \begin{cases} 1 & t_{\mathrm{on}} \leq t \leq t_{\mathrm{off}} \\ 0 & Otherwise \end{cases} \tag{2}$$

where $t$ is the time; $t_{\mathrm{on}}$ is the time stimulation is turned on and $t_{\mathrm{off}}$ is the time the stimulation ends.

To turn this step function into a softer response function, we introduce a second stimulus variable that increases depending on the step function $u(t)$, and has a built-in decrease proportional to the level of activation, i.e.:

$$\frac{d(Stimulus)}{dt} = k_{stim1} * u(t) - Stimulus * k_{stim2} \tag{3}$$

where $Stimulus$ is a state that represent the effect of the visual stimuli; $k_{stim1}$ describes the rate of stimulus activation and $k_{stim2}$ describes the rate of stimulus decay.

The electrical activity in the neurons triggers a metabolic response involving many different metabolic reactions. Herein, we consider a simplified network of metabolites and reactions, for which we have available measurements. More specifically, the simplified metabolic network described by this model consists of around a dozen reactions. These reactions represent the metabolic pathways in and directly around the tricarboxylic acid cycle (TCA cycle). At the top of this network, glucose diffuses across the blood brain barrier from blood vessels into the neuronal tissue ($Gluc_t$). The rate of this reaction is described by the variable $V_{Gluc}$, which is assumed to be irreversible and governed by mass action kinetics. Following the inflow, glucose is consumed and we here assume that this consumption occurs only via glycolysis. The rate of glycolysis ($V_{glycolysis}$) is assumed to have a basal rate and a Stimulus activated rate, both governed by mass action kinetics. With these assumptions, the ODE for $Gluc_t$ is given by:

$$\frac{d(Gluc_t)}{dt} = V_{Gluc} - V_{glycolysis} = k_{\mathrm{max_{gluc_c}}} * Gluc_c - k_{\mathrm{max_{gluc_t}}} * Gluc_t * (1 + Stimulus) \tag{4}$$

where $Gluc_t$ describes the concentration of glucose in the neuronal tissue and where the kinetic rate parameters for inflow and consumption of $Gluc_t$ are described by $k_{\mathrm{max_{gluc_c}}}$ and $k_{\mathrm{max_{gluc_t}}}$, respectively.

In the model, we assume that the plasma glucose concentration close to the activation site can be modified i.e., that $Gluc_c$ is a time-dependent state. In contrast, we assume a constant inflow of glucose to the local circulation since the visual stimulus is assumed to only impact the local metabolism.

$$\frac{d(Gluc_c)}{dt} = Glucose_{Blood} - V_{Gluc} = Glucose_{Blood} - k_{\mathrm{max_{gluc_c}}} * Gluc_c \tag{5}$$

where $Gluc_c$ describes the concentration of glucose in the local circulation, and where $Glucose_{blood}$ represents a constant inflow of glucose from the wider circulation.

Glucose in the tissue is metabolised in intracellular glycolysis which consists of several reactions. In this model, glycolysis is simplified into a single reaction that converts glucose to pyruvate. The rate expression for this is described by $V_{glycolysis}$, which was included in Eq (4). Pyruvate in turn is then either converted into lactate or sent to the TCA cycle. The rate

expression for conversion to lactate is described by $V_{LDH}$, which for simplicity is given by mass action kinetics. The steps leading to the TCA cycle can occur either via pyruvate carboxylase (PC) or via pyruvate dehydrogenase (PDH). The rate expression for PC is given by $V_{PC}$ and assumes Michaels-Menten kinetics. The second reaction going into the TCA cycle, PDH, also assumes Michaels-Menten kinetics but is given by a more complex rate expression, dependent on two substrates. The reason for this complexity is that the pyruvate that enters the TCA cycle via the PDH pathway is combined with oxaloacetic acid (OAA), which then, through several reactions, is turned into 2-oxoglutarate (OG). These multiple rection steps are in the model simplified into a single reaction (with reaction rate $V_{TCA1}$), which describes the first part of the TCA cycle (Fig 1C). The rate expression for $V_{TCA1}$ is thus dependent on two substrates and is assumed to have a saturation with respect to both Pyruvate and OAA. With all these reactions included the ODE for pyruvate becomes:

$$\frac{d(Pyr)}{dt} = V_{glycolysis} - V_{TCA1} - V_{LDH} - V_{PC}$$

$$= k_{\max_{gluc_t}} * Gluc_t * (1 + Stimulus) - \left( \frac{k_{max_{PO}} * Pyr}{K_{M_{Pyr}} + Pyr} * \frac{k_{max_{PO}} * OAA}{K_{M_{OAA}} + OAA} \right) - k_{max_{Pyr}} * Pyr$$

$$- \frac{k_{max_{pyr2}} * Pyr}{K_{M_{Pyr2}} + Pyr} \tag{6}$$

where $Pyr$ describe the concentrations of pyruvate; $k_{max_{PO}}$, $k_{max_{Pyr}}$, and $k_{max_{pyr2}}$ are kinetic rate parameters; and where $K_{M_{Pyr}}$, $K_{M_{OAA}}$ and $K_{M_{Pyr2}}$ are Michaelis-Menten constants. Note that $k_{max_{PO}}$, $k_{max_{Pyr2}}$ describe the maximum rate of $V_{TCA1}$ and $V_{PC}$, respectively, while $k_{max_{Pyr}}$ describes the rate of conversion from pyruvate to lactate.

Lactate is not considered in detail in this model and is simply described by a conversion from pyruvate ($V_{LDH}$) and a clearance term ($V_{clear1}$). The differential equation for lactate is given by:

$$\frac{d(Lac)}{dt} = V_{LDH} - V_{clear1} = k_{max_{Pyr}} * Pyr - k_1 * Lac \tag{7}$$

where $Lac$ is the concentration of lactate and $k_1$ is the rate parameter describing the degradation rate of lactate.

As described earlier, the model condenses the multiple reactions of the TCA-cycle that converts oxaloacetic acid (OAA) and pyruvate to 2-oxoglutarate (OG) into a single reaction TCA1 (reaction rate: $V_{TCA1}$). Further, the remaining reactions that returns OG to OAA is also simplified into a single reaction TCA2 (reaction rate: $V_{TCA2}$). Thus, the reactions TCA1 and TCA2 create a cyclical relationship between the model states of OAA and OG, which together make out the TCA cycle. Additionally, OAA can undergo transamination to form aspartate, this is described in the model as $V_{GOT}$. With these reactions, the full differential expression for OAA is thus described as:

$$\frac{d(OAA)}{dt} = V_{TCA2} + V_{PC} - V_{TCA1} - V_{GOT}$$

$$= k_{max_{OG1}} * OG + \frac{k_{max_{pyr2}} * Pyr}{K_{M_{Pyr2}} + Pyr} - k_{max_{PO}} * \left( \frac{Pyr}{K_{M_{Pyr}} + Pyr} * \frac{OAA}{K_{M_{OAA}} + OAA} \right) - k_{max_{OAA}} * OAA \tag{8}$$

where $OAA$ is the concentration of oxaloacetic acid in the tissue and where $k_{max_{OG1}}$, and $k_{max_{OAA}}$ are the kinetic rate parameters describing the reaction rates of $V_{TCA2}$ and $V_{GOT}$, respectively.

Analogously, OG is synthesised via TCA1 and is consumed by TCA2, but OG can also be transaminated to form glutamate. This reaction from OG to glutamate is denoted XM (reaction rate: $V_{xm}$). The full ODE for OG is thus given by:

$$\frac{d(OG)}{dt} = V_{TCA1} - V_{TCA2} - V_{xm}$$

$$= k_{max_{PO}} * \left( \frac{Pyr}{K_{M_{Pyr}} + Pyr} * \frac{OAA}{K_{M_{OAA}} + OAA} \right) - OG * \left( k_{max_{OG1}} + k_{max_{OG2}} \right) \tag{9}$$

where $OG$ is the concentration of 2-oxoglutarate, and $k_{max_{OG2}}$ is the kinetic rate parameter for the transamination of OG.

Lastly, the model describes the concentrations of three metabolites on the periphery of the TCA cycle that are vital to cerebral metabolism. The first of these metabolites is aspartate, which is converted from OAA via $V_{GOT}$. The subsequent metabolism of aspartate is implemented as a simple reaction denoted $V_{clear2}$

$$\frac{d(Asp)}{dt} = V_{GOT} - V_{clear2} = k_{max_{OAA}} * OAA - k_{max_{Asp}} * Asp \tag{10}$$

where $Asp$ is the concentration of aspartate and $k_{max_{Asp}}$ is the kinetic rate parameter describing the rate of further metabolization of aspartate, not described in detail by this model.

The second of the peripheral metabolites is glutamate, which as described is transaminased from OG. Glutamate is then converted into glutamine, the third peripheral metabolite, via glutamine synthetase (rate expression: $V_{GS}$). Reversely, glutamine is converted back to glutamate via phosphate-activated glutaminase (rate expression: $V_{PAG}$). Finally, glutamate is decreased by $V_{clear3}$, which is a simplified description for further metabolism and utilization of glutamate in the neuronal tissue. All these reactions and simplifications means that the ODE for glutamate is implemented as:

$$\frac{d(Glut)}{dt} = V_{xm} + V_{GS} - V_{PAG} - V_{clear3}$$

$$= k_{max_{OG2}} * OG + k_{max_{Gln}} * Gln - Glut * \left( k_{max_{Glut1}} + k_{max_{Glut2}} \right) \tag{11}$$

where $Glut$ is the concentrations of glutamate; $k_{max_{Gln}}$ and $k_{max_{Glut1}}$ are the kinetic rate parameters for the forward and reverse conversion of glutamine to glutamate, respectively; and $k_{max_{Glut2}}$ is the kinetic rate parameter of any further metabolism of glutamate, not described in detail in this work.

The last metabolite described in this model is glutamine, and the only glutamine dynamics described in this model is governed by the already described $V_{GS}$ and $V_{GLUT}$ rate expressions, i.e.

$$\frac{d(Gln)}{dt} = V_{PAG} - V_{GS} = k_{max_{Glut1}} * Glut - k_{max_{Gln}} * Gln \tag{12}$$

where Gln is the concentration of glutamine.

**2.3.2 Model structure for the NVC model.** The model structure for the NVC model was taken from the model presented by Sten *et al.* 2017 [23]. This model gives a detailed

physiological description of how neuronal activity affects the CBF, CBV, and the cerebral metabolic activity to generate a BOLD-response. For a detailed description and motivation of this model, please see the supplementary S1 Table or the original work [23], and all reactions in the complete model, along with simulation files are specified in the S1 Supporting Information file.

**2.3.3 Connection between NVC model and metabolism model.** The two models–the new metabolism model in 2.2.1 and the Sten *et al.* 2017 model–can be considered as two separate standalone models. However, we here also present and analyse a connected version. In this connected version, Eqs (3), (4), and (5) were removed from the metabolic model since corresponding reactions already exist in the Sten *et al.* 2017 model. Further, the reaction named $V_{glycolysis}$ in Eq (6) was replaced with the reactions baseMet (reaction rate: $V_{baseMet}$) and stimMet (reaction rate: $V_{stimMet}$), which are a part of the Sten *et al.* 2017 model [23]. The first of these reactions, baseMet describes the basal non-stimulated metabolism of glucose and oxygen, and stimMet describes the increase in metabolism caused by neuronal activity. With these alterations implemented, the ODE for pyruvate is described by:

$$\frac{d(Pyr)}{dt} = V_{baseMet} + V_{stimMet} - V_{TCA1} - V_{LDH} - V_{PC}$$

$$= Glucose_A * (k_{basalMet} * O_{2A}^{k_{prop1}} + Delay_M * O_{2A}^{k_{prop2}}) - \left(\frac{k_{maxPO} * Pyr}{K_{M_{Pyr}} + Pyr} * \frac{k_{maxPO} * OAA}{K_{M_{OAA}} + OAA}\right) - k_{max_{Pyr}}$$

$$* Pyr - \frac{k_{max_{pyr2}} * Pyr}{K_{M_{Pyr2}} + Pyr} \tag{13}$$

where $k_{basalMet}$, $k_{prop1}$ and $k_{prop2}$ are kinetic rate parameters; where $Glucose_A$ and $O_{2A}$ are the respective concentrations of glucose and oxygen in the neuronal tissue; where $Delay_M$ represents an intermediate state of the stimulus effect on the metabolism. The variable $Glucose_A$ describes the same quantity as the state $Gluc_t$ described in the removed Eq (4). The reason for this change of name in the combined model is that $Glucose_A$ was an already implemented state in the NVC model. The parameters and states of the NVC model are described in further detail in Sten *et al.* 2017 [23], and both that and the combined models are described in the supplementary S1 Table.

In this combined model, as in the two constituent sub-models, the observed model properties, or model observables, consist of four metabolic states: *Lac*, *Glut*, *Gluc$_t$*, and *Asp*. Additionally, a fifth model observable is the BOLD-signal defined in Sten *et al.* 2017 as a signal proportional to the amount of deoxy haemoglobin (*dHb*). The BOLD-signal is formally defined as:

$$BOLD = e^{-ky*dHb} \tag{14}$$

where *dHb* is the amount of deoxy haemoglobin and *ky* is a scaling parameter.

More specifically, each of the five model observables are defined as a percentage difference from a steady-state baseline. In practice, this is achieved by dividing the observable value with its corresponding steady-state value, withdrawing one and multiplying with 100%:

$$\hat{y}_i = ky_i * \left(\frac{X_i}{ssX_i} - 1\right) * 100\% \tag{15}$$

where $X_i$ is one of the observable properties *Lac*, *Glut*, *Gluc$_t$*, *Asp*, or BOLD; where $ssX_i$ is the corresponding steady-state value, and where $ky_i$ is the i$^{th}$ scaling parameter.

## 2.4. Model evaluation

**2.4.1. Parameter estimation.**　The model parameters ($\theta$) are estimated from experimental data ($D$), by minimizing the negative log-likelihood function $J$:

$$J(\theta) = -\log(L(D|\theta)) = \frac{1}{2} \sum_{e=1}^{n_e} \sum_{o=1}^{n_o^e} \sum_{s=1}^{n_s^{e,o}} \left[ \log(2\pi(\sigma_s^{e,o})^2 + \left( \frac{y_s^{e,o} - \hat{y}_s^{e,o}(\theta)}{\sigma_s^{e,o}} \right)^2 \right] \quad (16)$$

where $\theta$ is the model parameters; $n_e$ is the number of experiments $e$, $n_o^e$ is the number of observations in experiment $e$; $n_s^{e,o}$ is the number of samples at observation $o$ in experiment $e$; $y_s^{e,o}$ is the measured data sample point; $\hat{y}_s^{e,o}(\theta)$ is the model simulation for the corresponding sample point; and $\sigma_s^{e,o}$ is the standard deviation of the sample point.

The function $J(\theta)$ is implemented as the objective function for a suitable optimization algorithm (see section 2.6) and optimized with respect to the parameters $\theta$.

**2.4.2 Identifiability analysis.**　For the model presented herein two types of identifiability analysis were performed. A structural identifiability analysis to evaluate if the model structure resulted in any unidentifiable model states or parameters, and a practical identifiability analysis to evaluate what simulated model states and parameters can be accurately estimated from the experimental data. For the structural identifiability analysis, a local identifiability analysis was performed on the metabolism model described in section 2.3.1. An implementation of the recently developed methods described by Thompson *et al.*, 2022 [40] was used. These methods extend the algorithm presented by Sedoglavic *et al.* in 2002 [41] to determine local identifiability of states and parameters when different numbers of derivatives of signals are available. This analysis showed that all model states and parameters were identifiable when at least eight derivatives of the measurement signals was available. A complete report of these results is presented in the supplementary S2 Table.

For the practical identifiability analysis, the uncertainties of the simulated model states and the parameters were estimated through a Markov chain Monte Carlo (MCMC) sampling approach. A posterior distribution of the parameter values was generated, using $10^5$ samples, and all parameter sets found acceptable according to a $\chi^2$-test was collected. The uncertainty is then determined by the confidence interval:

$$CI_{\alpha,df} = (J(\theta) \leq J(\theta^*) + \Delta_\alpha(\chi^2_{DoF})) \quad (17)$$

Where $\alpha$ is the confidence level; $\Delta_\alpha(\chi^2)$ is the $\alpha$ quantile of the $\chi^2$ statistic [42,43]; $DoF$ is the degrees of freedom; and $\theta^*$ are the optimal parameters. In this work, the $DoF$ is equal to the number of model parameters (n = 58). The optimal parameter values and the boundaries of the parameter distributions can be found in the supplementary S3 Table.

## 2.5 Experimental data

In this work, experimental data from previously published studies have been used for model evaluation. Mainly, H-MRS data from human subjects exposed to visual stimuli have been considered for the model evaluation presented herein. Here follows a condensed description of the data, for further details please see the original publications for each dataset. When necessary, data were extracted from publication figures, using the WebPlotDigitizer tool [44].

**2.5.1 Metabolic estimation data.**　In their work, Lin *et al.* 2012 [34] present metabolite data gathered from the visual cortex of human subjects using functional MRS. Data was gathered form ten healthy subjects (7 males, 3 females, 25±3 years old) and the data used herein consists of two sets of four metabolites. The first data set describes the relative change in metabolic concentrations in the primary visual cortex for lactate, glutamate, glucose, and aspartate,

for a single visual stimulus. The second data set describes the relative change in metabolic concentrations for the same four metabolites but for a double visual stimuli paradigm. The visual stimulus consisted of contrast-defined wedges, moving towards and from a central point. The single stimulus paradigm consisted of a 6.6-minute baseline followed by a 13.2-minute exposure to the visual stimulus, followed in turn by a 19.8-minute recovery. In the double stimulus paradigm, the baseline was established for 9.9 minutes followed by a two 9.9-minute periods of exposure to the visual stimulus, with a 9.9-minute rest period in between.

**2.5.2 BOLD estimation data.** The data used to train the BOLD-response is gathered from Witt *et al*. 2016 [45] and consists of fMRI data from a visual-motor task. The data was gathered from 11 healthy subjects (5 males, 6 females, 21–28 years of age) that was tasked to push a button in response to a certain visual stimulus. The visual stimulus was shown for 0.5 seconds, and a BOLD-response was extracted from each subject. The positive BOLD responses were gathered from the bilateral primary cortex, while negative BOLD-responses were gathered from the posterior cingulate cortex. The BOLD data is normalized as a percentage signal change.

**2.5.3 Metabolic validation data.** The model presented herein was validated with respect to independent validation data *i.e*., data that was not been used for model training. This data was taken from two independent studies. The first such study is the work presented by Schaller *et al*. 2013 [36]. In this work, ten healthy subjects (9 males, 1 female, 20–28 years of age) were exposed to a visual stimuli and changes in metabolic concentrations for lactate, glutamate was gathered via MRS. The data was sampled at 75 time points over a 25-minute period. This period consists of a 5-minute baseline period followed by two periods of exposure to visual stimuli for 5-minutes each. These exposure periods were interwoven by a 5-minute rest period and followed by a 5-minute recovery period.

The second study from which validation data were gathered is the work presented by Bednařík et al. 2015 [35]. In this work, the authors present MRS data that shows how metabolic concentrations of glutamate, aspartate, lactate, and glucose change as a response to a visual stimulus. The data is gathered form fifteen healthy subjects (7 males, 8 females, 33±13 years of age). The visual stimulus consisted of a red and black flickering checkerboard. For the metabolite measurements the stimulation sequence consisted of a 5.3-minute baseline acquisition, followed by a stimulation-rest-stimulation-rest sequence were each block lasted 5.3 minutes. The metabolic concentration data was sampled with a 2.7-minute resolution, for a total of nine sample points for each metabolite. For the model validation the model was used to simulate the metabolic response that corresponded to the experimental setting of the validation data, described above. The parameters used for this simulation were the optimal parameters, with respect to the estimation data, obtained from the parameter estimation described in Section 2.4.1.

## 2.6 Model implementation

Model implementations and simulations were done in the "Advanced Multi-language Interface to CVODES and IDAS" (AMICI) toolbox [46–48]. The implementation of the equations described in section 2.3 and the subsequent model analysis were done in MATLAB by The MathWorks, Inc. releases 2017b and 2020a. Model parameter estimation were done using the "Metaheuristics for systems biology and bioinformatics global optimization" (MEIGO) toolbox [49] with the enhanced scatter search (ess) algorithm.

## 3 Results

We have developed a new model for the central metabolism in the visual cortex and connected this model to our existing model for the NVC [23]. The model is described in detail in

Materials and Methods and all simulation files are available in the supplementary information (S1 Supporting Information). The model has been developed using experimental data from two different studies, Lin *et al.* 2012 [34] (Fig 2) and Sten *et al.* 2017[23] (Fig 3). This data was used to estimate the model parameters. Additionally, the model has been evaluated with experimental data gathered from Schaller *et al.* 2013 [36] (Fig 4) and Bednařík *et al.* 2015 [35] (Fig 5). This data was used as independent validation data to assess the model's ability to explain data for which it had no prior knowledge. This provides some support for the included mechanisms and predictive capability of the model. Finally, we demonstrate the usefulness of the model by showing how one can provide reasonable simulation results of metabolism in experiments where metabolism has not been measured. In these simulations, we also used the model to predict how long a stimulation would need to be to generate a measurable metabolic response (Fig 6).

## 3.1 Metabolic estimation results

As mentioned, part of the dataset used to train the model was gathered from Lin *et al.* 2012 [34], described in further detail in Section 2.5.1. Briefly, this data describes the relative change in metabolic concentrations in response to either a single stimulus (Fig 2A–2D, error bars) or a double stimulus (Fig 2E–2H, error bars). More specifically, the single stimulation consisted of a sustained visual stimulus for 13.6 minutes (Fig 2A–2D, black bar). In contrast, the double stimuli paradigm consists of two periods of sustained visual stimuli for 9.9 minutes, separated by a 9.9-minute rest period (Fig 2E–2D, black bars). The experimental data is presented as mean ± SEM (Fig 2, red error bars).

The model was simultaneously fitted to both the metabolic data in response to the single and double stimulation paradigm in Lin *et al.* 2012 [34], as well as to the BOLD data in Sten *et al.* [23] (presented in Section 3.2). The resulting agreement between data (error bars) and the simulation (shaded areas) is shown in Figs 2 and 3. As can be seen, for the single stimulus, the model clearly describes the increase in lactate (Fig 2A) and glutamate (Fig 2B) and the decrease in glucose (Fig 2C) and aspartate (Fig 2D), in response to the visual stimuli. Similarly, the double stimuli paradigm shows a model behaviour that is consistent with the experimental data (Fig 2E–2H): during the visual stimulation the lactate and glucose levels increase (Fig 2E and 2F) and the glucose and aspartate levels decrease (Fig 2G and 2H), relative to their baseline. These visual assessments are also supported by a statistical $\chi^2$-test ($J(\theta^*)$ = 213.44, threshold: $\chi^2(\alpha = 0.05, DoF = 191)$ = 224.24) (Materials and Methods).

## 3.2 The model simultaneously agrees with BOLD data

The data used to train the model's BOLD-response originates from Witt *et al.* [45] and was used to train the model in Sten *et al.* [23]. The data consists of a positive BOLD-response (Fig 3A) and a negative BOLD-response (Fig 3B). This data is described in more detail in Section 2.5.2. The stimulus used is a short (0.5 s) visual stimuli at $t$ = 0.1 (Fig 3, black line), and the data is presented as a percental mean ± SEM deviation from the baseline (Fig 3, red error bars). As described in the previous section, the model was simultaneously fitted to both these BOLD data and the MRS data (Fig 2). The agreement between the BOLD data and simulation (blue shaded areas) is presented in Fig 3. As can be seen, the model displays all the expected features of an archetypical BOLD response: a small initial dip around 2 seconds after the stimulation, a larger main response at around 5–7 seconds and a post-peak undershoot between around 10–16 seconds [22,23]. This response is in good agreement with the data, which again is formally supported by a $\chi^2$-test ($J(\theta^*)$ = 213.44, threshold: $\chi^2(\alpha = 0.05, DoF = 191)$ = 224.24).

### 3.3 Model validation against additional MRS data

Once the model had been trained on the MRS data gathered from Lin *et al.* 2012 [34] and BOLD data from Sten *et al.* [23], the model was validated against the data gathered from Schaller *et al.* [36] (Fig 4) and Bednařík *et al.* [35] (Fig 5). The MRS data gathered from Schaller *et al.* describes the change in metabolic concentrations of lactate and glutamate in response to two visual stimuli, with a short break in between. The data is herein presented as a relative change in mean concentration with the SEM displayed as error bars (Fig 4, red error bars). The model predicted dynamics for the concentrations of lactate (Fig 4A) and glutamate (Fig 4B) which are displayed as the blue shaded area, and which is in very good agreement with the data. For changes in lactate concentration, the model provides a good prediction of how lactate increases as a response to the visual stimulus i.e., the same increase can be seen in the data (Fig 4A). For the changes in glutamate, the model predicts a slight increase when the visual stimulus is turned on. This is in accordance with the data. However, when the visual stimulus is turned off (at 600 s and 1200 s) the model describes a decrease in glutamate levels, but this decrease is slower than the corresponding decrease seen in the data. This makes the model prediction slightly too high for all points following the first stimulus period (Fig 4B). All in all, given that the model is not trained for any of these data, the agreement between model and data is good, both qualitatively and quantitative, and comparable to the size of the experimental variations and uncertainties.

Further, the model was simultaneously validated using the data gathered from Bednařík *et al.* [35]. This data also describes changes in metabolic concentrations as a response to a sequence of visual stimuli, further described in Section 2.5.3. This data set contains data for lactate (Fig 5A), glutamate (Fig 5B), glucose (Fig 5C), and aspartate (Fig 5D), and again the data is presented as the mean concentrations ± SEM indicated by the red error bars. Once again, the model predicts the changes in metabolic concentrations with good accuracy (Fig 5, blue shaded areas). Similarly, to previous results, the model predictions for lactate and glutamate show an increase when the visual stimulus is turned on and as previously this is supported by the data (Fig 5A and 5B). Also, similarly to before, the predicted recovery rate for glutamate appears slightly too slow when compared to data (Fig 5B). Further, for glucose and aspartate, the model-predicted levels are slightly lower than what is indicated by the data. For glucose, the model prediction is consistently below the data points after the first period of stimulus (Fig 5C). For aspartate, the model prediction is consistently below the data points following the start of the first stimulus period (Fig 5D). All in all, the overall agreement between model predictions and validation data is qualitatively correct, and quantitatively the simulation-data differences are comparable to the experimental uncertainty, at least for lactate, glutamate, and glucose.

### 3.4 Using the final model: simulating non-measured variables

Once the model had been validated, we used the model to predict various variables and conditions for which we have no data. In particular, we investigated how long a visual stimulation needs to be to generate a measurable response in metabolic concentrations. This was achieved by evaluating the amplitude of the metabolic responses for glucose, lactate, aspartate, and glutamate, for different stimulation lengths (Fig 6A, grey shaded areas). The length of the stimulations was varied from 0.5 s to 10 000 s. As to be expected, the amplitude of the metabolic response increases as the length of the stimulation increases. Furthermore, our simulations suggests that the metabolic response reaches a maximal amplitude, after a few hundred seconds. For lactate and aspartate, a stimulation of around 200 s or longer yields this maximal response, for glutamate, the maximal response is reached for stimulations around 300 s, and

for glucose the stimulation length needs to be around 600 s to reach a maximal response (Fig 6A).

Further, the model can be used to make predictions for non-measured variables in an existing experiment. In Lundengård *et al.* [22], there are BOLD response data for a series of two short 0.5 seconds stimuli, with 4 seconds in-between, but no metabolic data were collected. Our model can be used to assess what the metabolism probably looked like. The model can accurately predict this BOLD response (Fig 6B), but it also predicts the corresponding metabolic response for the same stimulation paradigm. These predictions shows that the metabolic response is slower than the BOLD response: the BOLD signal is completely over after 15–20 seconds, even though the metabolism still is maximal (Fig 6C–6F). Furthermore, for such short stimuli, the amplitude of the metabolic response is considerably lower than in the previous examples above (Figs 2, 4, and 5). This lower amplitude is consistent with the results in Fig 6A. This means that corresponding metabolic changes would not be experimentally detected, which makes sense since metabolic experiments usually have longer paradigms. The simulations, however, provides an estimation of MRS experiments with higher signal-to-noise-ratio.

## 4 Discussion

Mathematical modelling provides a great potential to understand the complex interplay between cerebral metabolic, hemodynamic, and neuronal activity. Herein, we present a mathematical model that can explain previously unmodelled MRS data for the central cerebral metabolism. Further, we present a first connection between a minimal metabolic model and a mechanistically detailed NVC model. We show that this combined model can accurately describe MRS and fMRI data simultaneously and that the model can also describe independent validation data, not used for model training. This MRS data, used to train the model, consisted of time series that show how concentrations for glucose, lactate, aspartate, and glutamate change as a response to visual stimuli (Fig 2). The fMRI data consisted of a positive and negative BOLD-response (Fig 3), also from a visual stimulus. Similarly, the independent MRS data, used for model validation, also consists of time-series that show the changes in metabolic concentrations as a response to visual stimuli (Figs 4 and 5). Finally, we show that this model can be used to make predictions of non-measured variables and conditions, and that it can be used to assess *e.g.*, how long a stimulation needs to be to generate a measurable change in metabolic concentrations, and what the likely shape of the metabolism is in cases of shorter stimulations, when MRS cannot detect any signal (Fig 6).

In this study we have based our modelling of the cerebral metabolism on the MRS data gathered by Lin *et al.* 2012 [34], Schaller *et al.* [36], and Bednařík *et al.* [35]. However, MRS is by no means the only measuring technique employed to investigate the cerebral metabolic pathways. One more accurate, but also more demanding, approach to determine cerebral metabolic pathways is modelling combined with $^{13}C$ MRS data. In this technique researchers insert isotopically labelled substrates into the metabolic system and measure the fractional enrichment of these isotopes in metabolic products [50,51]. Mathematical modelling based on this type of data then allows for accurate quantification of different metabolic pathways during different conditions e.g., before and after stimulation [52]. However, these models require specific knowledge regarding the relevant atom transitions of the isotopically labelled substance. This means that such atom-based models are more difficult to scale up, and that they mainly are used to investigate specific metabolic pathways in isolation. Furthermore, such methods are mostly applied to animal and *in vitro* experimental system.

In this study we also investigated how the model predicts that the length of the stimulations affects the metabolic response. Most studies that investigate the cerebral metabolism via MRS

uses very long periods of visual stimulation. For instance, the study from which we have gathered data [34] uses a period of continuous visual stimuli that lasts almost ten minutes. While our model suggests that this is the time required for the maximal signal amplitude and thus the largest signal-to-noise ratio (SNR), our simulations also suggest that a detectable signal-to-noise ratio could be reached with shorter stimulations, such as 2 minutes (Fig 6A). However, it should be noted that such shorter stimulation protocols still need a longer data-collection period, since the dynamics of the metabolism is slow, and probably continues long after short stimulations are over (Fig 6). These statements are based on our interconnected model that can be used to make predictions of probable behaviours of variables in cases where data is not available. Here, this new possibility was illustrated by doing model predictions for the metabolic responses, in an experimental scenario where only BOLD responses were measured (Fig 6). This shows that two 0.5 s stimulations with 4 s in-between will give rise to two, in principles, measurable peaks within 10 s in the BOLD signal, but to slower and lower responses in the metabolites: <5% response in glucose and lactate, and <2% response in glutamate and aspartate. These expected changes are smaller than the usual experimental variability (Figs 2, 4, and 5), implying that those predictions would probably not be experimentally detectable. However, these predictions of short stimulation metabolic response are done simultaneously with a prediction of the BOLD-response (Fig 6B) which is validated by experimental data which increases the confidence that the prediction for the metabolic response is also reasonable. This likelihood is of course also supported by the fact that the model can both describe and predict other types of metabolic data. This shows that we can do a prediction of a situation that likely cannot be experimentally measured. As such, there is potential for our model to be used as a tool for experiment-design of an experiment that has shorter stimulations than the original studies, but that still would give measurable responses.

There are a couple of limitations regarding this work that should be mentioned. As is the case with all mathematical models, the model presented herein is the result of several simplifications and assumptions. For instance, several previous works have shown that aspects such as compartmentalisation between neuronal and glial tissue and glycogen metabolism have important roles in the cerebral metabolism [26–29,32,52,53]. However, adding additional complexity to the model, for instance in the form of multiple compartments, does not necessarily improve the model's ability to explain the data we have considered in the manuscript, but it does reduce the model's parameter identifiability. This means that the estimated confidence intervals for the parameter values might increase by an order of magnitude, or more, by adding complexity to the model. This is illustrated by an example case presented in supplementary file S1 Text, where the parameter identifiability of a two-compartment version of the model presented herein is analyzed. Consequently, the model gives less accurate estimates of the parameters that are included while not improving the explanation of the given experimental data. The reason for this reduced identifiability is that the complexity of the model will increase beyond the complexity of the experimental data. For instance, the data we used to train the model (Lin *et al.*, 2012) [34]cannot distinguish which compartment the change in a metabolite's concentration occurs in. Similarly, adding additional metabolic pathways such as an inflow of glycogen to the model would make the production of glucose less identifiable i.e., more uncertain. Therefore, the source of glucose could be either from the blood or from glycogen degradation. Given the considered data it would not be possible to specify the source of glucose. For these reasons we choose to maintain the minimal model approach even though this means that the model does not include certain previously modelled aspects of the cerebral metabolism.

Another such simplification is the current interpretation of the oxygen consumption as implemented in Sten et al. 2017 [23]. More specifically, this implementation means that the

only part of the metabolism that affects the $CMRO_2$ is the glycolysis step *i.e.*, the conversion form glucose to pyruvate as described by Eq (13). This interpretation is a simplification, as oxygen is consumed both in the glycolysis and in the mitochondrial oxidative phosphorylation [54]. However, the effect of oxygen in the glycolysis seen in Eq (13) has no corresponding effect on the steps that represents the TCA-cycle ($V_{TCA1}$ and $V_{TCA2}$ in e.g., Eq (8)). Such an effect has not been necessary to accurately describe the data presented in this work. The inclusion of a more detailed description of the oxygen metabolism would produce a more physiologically accurate interpretation of the cerebral metabolism but also would require more data than we had access to in this work.

Finally, the metabolic model presented herein makes simplifications with respect to the units of the different states defined in the model. The metabolic concentrations have been modelled in arbitrary units and are scaled by unknown scaling parameters as defined in Eq (15). Despite these simplifications, the model is still able to accurately explain both the metabolic and hemodynamic dynamic activity that is seen in experimental data, as well as predict independent validation data.

Furthermore, the presented model uses a simplistic expression for calculating the BOLD-signal (see Eq 14). This simple expression is taken directly from the Sten *et al.* 2017 [23] model and does not directly take into account the effects of variables such as CBV, CBF and $CMRO_2$. However, these variables do indirectly affect the calculated BOLD-signal as they are incorporated into the expression for dHb. There are models that gives a more detailed description of the BOLD-signal; for instance, Havlicek *et.al.* 2015 [55] present an expression where the BOLD signal constitutes a combination of the extravascular and intravascular signals. Further, one other more detailed version of calculating the BOLD signal, not included if in our model, is to consider the compartmentalization of the intravascular component as is done in e.g. Griffeth and Buxton 2011 [56] and Kim and Ress 2016 [57]. In these papers, the BOLD signal is calculated as a sum of contributions from arterial, capillary, and venous vessels, as well as an extravascular compartment. A more comprehensive implementation of the BOLD-signal would likely lead to a model that could more accurately describe a versatile set of BOLD-responses possibly improving the prediction of the BOLD-response seen in Fig 6B, to be more in line with the experimental data. Such an implementation is included in Sten *et al.* 2021 [25] where multiple sets of experimental data are used to identify key mechanisms of the BOLD-response that are preserved across different species and time-scales. The reason why such an expression for the BOLD -signal was not implemented in this work is that the model developed herein focuses on the metabolic responses and the connection to the already existing Sten *et al.* 2017 NVC model.

Our new combined NVC-metabolism model opens the door to several important applications in the future. First, one such application is model-based experiment-design to *e.g.*, design new shorter stimulation paradigms, as discussed above. Second, another application is to integrate a wider variety of data into the same model-based analysis. In other words, this model could incorporate data from future experiments where e.g., both BOLD and metabolism has been measured simultaneously. Third, an extension of this application is to incorporate data for CBV and/or CBF. Since all these variables—BOLD, CBV, CBF, and metabolism—are interconnected, they should be analysed using one and the same model, to fully exploit all the information that is contained in such a joint dataset. However, for that to be possible, a more advanced interconnected model than the one presented herein is needed, since this model probably does not provide a good enough description of CBV and CBF dynamics. Nevertheless, once such an interconnected model is in place, it can be used to provide a new updated ability to use a model to infer $CMRO_2$, compared to the highly simple model that is used presently [56]. Fourth, such integrated analyses could also be useful in clinical contexts, since

multiple variables analysed together would provide an integrated understanding of the brain, with the potential for new biomarkers and mechanistic insights regarding patient screening, stratification, and monitoring [58]. Finally, a future metabolic-NVC model can also be used in multi-organ digital twin models, that also can be used for a variety of applications. All in all, our new integrated model is the first to include intracellular details regarding both metabolism and NVC (Fig 1A), and even though it still is just a first step in this new direction, it points to way towards many important applications in the future.

## Supporting information

**S1 Fig. An illustration of the two-compartment version of the model presented in Fig 1C.** Illustrating the division of the metabolic reactions into a glial and a neutron compartment with some cross over reactions.
(TIF)

**S2 Fig. Comparison of parameter distributions from the identifiability analysis.** The box chart illustrates the parameter distributions for four parameters ($k_{max_{pyr2}}$, $k_{M_{Pyr2}}$, $K_{max_{Glut1}}$, $k_{max_{Glut2}}$). With the distribution of the parameters in the single compartment model shown in blue and distributions of the corresponding parameters for the two-compartment model shown in red and yellow respectively.
(TIF)

**S1 Table. A complete list of the implemented model equations form the Sten et al.** 2017 model [23]. The equations are taken with the consent of the authors of the original work and are presented here as for the convenience of the reader. Please see the original publication for a detailed description of the model equations.
(DOCX)

**S2 Table. The results from the structural identifiability analysis that shows how many derivatives of the measurement equations are sufficient for each model state and parameter to be identifiable.** The calculations are based on an implementation of the methods presented by Thompson et al., 2022 [40].
(DOCX)

**S3 Table. A list of the parameter values for the presented model.** The parameter values correspond to the values obtained for the best fit to the estimation data. The upper and lower bounds correspond to the range of acceptable parameter values obtained from the uncertainty analysis.
(DOCX)

**S1 Text. An example case of the problems with parameter identifiability.** This example illustrates the problems with parameter identifiability that arise when additional model complexity is introduced. In this example a two-compartment version of the metabolic model presented here is fitted to the data for the metabolic responses and the resulting parameter identifiability is analysed.
(DOCX)

**S1 Supporting Information. The zip-file that contains all the necessary files required to reproduce the results presented in this paper.** To ensure compatibility please ensure that all the requirements described in the provided read me file are fulfilled.
(ZIP)

## Acknowledgments

Preliminary analysis leading up to these results presented herein have been done by Sofie Johansson, who worked on this project in a student internship.

## Author Contributions

**Conceptualization:** Sebastian Sten, Maria Engström.

**Data curation:** Nicolas Sundqvist, Sebastian Sten, Maria Engström.

**Formal analysis:** Nicolas Sundqvist, Sebastian Sten, Peter Thompson, Benjamin Jan Andersson.

**Funding acquisition:** Maria Engström, Gunnar Cedersund.

**Investigation:** Nicolas Sundqvist, Sebastian Sten, Gunnar Cedersund.

**Methodology:** Nicolas Sundqvist, Sebastian Sten, Peter Thompson, Benjamin Jan Andersson, Maria Engström, Gunnar Cedersund.

**Project administration:** Maria Engström, Gunnar Cedersund.

**Resources:** Nicolas Sundqvist, Sebastian Sten, Maria Engström, Gunnar Cedersund.

**Software:** Nicolas Sundqvist, Sebastian Sten, Peter Thompson.

**Supervision:** Sebastian Sten, Maria Engström, Gunnar Cedersund.

**Validation:** Nicolas Sundqvist.

**Visualization:** Nicolas Sundqvist.

**Writing – original draft:** Nicolas Sundqvist, Gunnar Cedersund.

**Writing – review & editing:** Sebastian Sten, Peter Thompson, Maria Engström, Gunnar Cedersund.

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
