## [Decision Letter · Decision Letter 0]

22 Jun 2022

Dear Mr Sundqvist,

Thank you very much for submitting your manuscript "Mechanistic model for human brain metabolism and its connection to the neurovascular coupling." for consideration at PLOS Computational Biology.

As with all papers reviewed by the journal, your manuscript was reviewed by members of the editorial board and by several independent reviewers. In light of the reviews (below this email), we would like to invite the resubmission of a significantly-revised version that takes into account the reviewers' comments.

We cannot make any decision about publication until we have seen the revised manuscript and your response to the reviewers' comments. Your revised manuscript is also likely to be sent to reviewers for further evaluation.

Sincerely,

Anders Wallqvist

Associate Editor

PLOS Computational Biology

Daniel Beard

Deputy Editor

PLOS Computational Biology

Reviewer's Responses to Questions

**Comments to the Authors:**

Reviewer #1: The authors integrated a simple computational model of brain energy metabolism into previously published models of neurovascular coupling.

My general consideration is that the fact that a model can fit some experimental data is not enough. Being able to get a good fitting of the data using a parameter optimization procedure is not a surprise, because model parameters are always many more than experimental measurements (i.e., the system is under-determined). What is important is to compare the optimized parameters with experimental equivalents. But in order to do this, we need that each process (and by extension, each parameter) has a clear physiological significance. In the present model, the correspondence between model equation and physiology is lacking.

For instance, the authors aim to fit experimental 1H-MRS data for brain aspartate and glutamate concentration changes observed in response to sensory stimulation. These experimental findings have been interpreted by current literature as reflecting a complex dynamics of aspartate and glutamate across mitochondria within the so-called malate-aspartate shuttle (MAS), which includes (at least) two transaminations and two carrier-mediated co-transport processes. Well, in this paper there is nothing about the MAS, which implies that any parameter that results from the optimization procedure cannot be interpreted using current knowledge.

Let's make a simple example. The authors consider a generic glutamate-oxalacetate transaminase (GOT) maximal reaction rate (kmax-OAA in Eqs. 8 and 10; or Vmax_OAA1 in Supplementary Table 2) with an optimized value of 0.0007 s-1. How we compare this value (which is a result) with current literature? The present model lacks compartments, so we don't know whether it is the cytosolic or the mitochondrial reaction. The modeled reaction (OAA -> ASP) doesn't even include the full stoichiometry (OAA + GLU <-> OG + ASP), but this is (wrongly) splitted in two, with the other term (OG -> GLU) having a maximal rate (kmaxOG2in Eqs. 9 and 11; Vmax_OG2 in Table S2) of 0.0184 s-1. Experimental values for isolated cerebral mitochondrial GOT (Ferrari et al, J Neurochem 2018) are around 0.4 umol/min/mgprotein, which should roughly be 0.006 s-1 (assuming an aspartate concentration of 1 mmol/L and 1 mgprotein/ml in mitochondria). This number is substantially different from both the above-mentioned optimized values used in this model.

For these reasons, in this particular review, I did not examine the Results and the Discussion sections, because I have quite some concerns on the modeling approach. Once these are solved, I'll be happy to provide further feedback.

Main concerns

- The very premise (abstract and introduction) that there is a lack of mathematical models integrating vascular and metabolic responses to stimulation is not entirely correct. There were published papers well before the first one published by the authors (Lundengård et al., PLoS Comput Biol 2016) that attempted to do exactly that (although with different degree of accuracy in mechanistic details and possibly different scientific questions), including but not limited to De Pitta et al., J Biol Phys 2009, Mangia et al., J Neurochem 2009, and DiNuzzo et al. J Neurophysiol 2011. I guess there is much more.

- Contrary to already published models of brain activity/metabolism/blood flow, the present model does not include compartmentalized energy metabolism (e.g., neurons and astrocytes). Neither it includes glycogen metabolism. Neither it includes the main enzyme-catalyzed reactions (e.g., glycolysis). Neither it includes the main transport reactions (e.g., glucose and lactate transport across the blood-brain barrier). Many model rate equations are based on simple mass-action kinetics, which is wrong for many reaction/transport processes. For example, glucose transport (equation 4) in the brain is described by reversible Michaelis-Menten kinetics that has well-established parameters (see Simpson et al., JCBFM 2007). Besides, several reactions are lumped together (e.g., glycolysis), which doesn't allow to account for: (i) redox balance, (ii) allosteric modulation (e.g., PFK), and (iii) product-inhibition (e.g., HK). Why did the authors not use prior knowledge?

- Last but not least, I am not fully convinced of the iterative approach used to validate the model (Section 2.2, Page 3). Specifically, even in the circumstance that some experimental data cannot be fitted using a specific reductionistic model, it doesn't mean that this model is necessarily wrong and has to be discarded. Indeed, the addition of new components might introduce nonlinear behaviors that can explain observations. Do the authors have any reference to a methodology paper that implements such a model construction strategy and rigorously examine the consequences?

Few other specific points

- Page 1 Line 12 (abstract): Strictly speaking the neurovascular coupling connects cerebral activity and blood flow, while neurometabolic coupling connect cerebral activity and glucose/oxygen consumption. The same applies to Page 2 Lines 37-38 (introduction).

- Page 2 Line 32 (introduction): I would include examples (e.g., glucose in the form of glycogen, and oxygen in globins). Yet, while oxygen storage is very limited, glucose storage as glycogen is quite substantial (Oz et al, Neurochem Res 2017).

- Page 4 Line 115 (Section 2.3.1): Why it has to be a visual stimulation? It doesn't seem that there is any particular detail in the stimulation equations 2 and 3 that is specific for the visual system, isn't it? It could well be a cognitive stimulation, too. Please justify or remove "visual". The same applies to the Results section. How is this model specific for the visual cortex?

- Page 4 Eq.3: Why the decay is soft (exponential), while the onset is not? Normally, in the literature stimulation functions consisting of a constant term plus a gamma function are used.

- When using Michaelis-Menten kinetics, I suggest referring to maximal reaction rate as Vmax (this is the standard), while k is commonly used for mass-action kinetics.

- Page 5 equation 5: Please use "blood" not "body" for subscript to glucose.

- Page 5 equation 6: The reaction rate equation VTCA1 is conceptually wrong (multiplication of two MM equations). By the way, the whole process (VTCA1-VTCA2) doesn't take into account carbon dioxide (so some mass is missing).

- Page 5 equation 7: I suggest replacing "lactate degradation" (ambiguous and possibly wrong) with "lactate clearance" (i.e., transport out of the brain tissue via blood or lymphatics), which is a physiological mechanism.

- Page 6 equation 10: What physiological process is responsible for "degradation" of aspartate?

- Page 7 Line 197: GLUT is commonly used for "glucose transporter". I suggest using PAG for glutaminase (i.e., phosphate-activated glutaminase).

- Supplementary Table 2: please check out for optimized values too close to boundaries (e.g., ky2). By the way, please use the same identifiers (only one for each process) in the main manuscript text and in the Table.

- It would be nice to perform a sensitivity analysis after optimization.

Reviewer #2: Vibe authors propose an interesting model describing neurometabolic coupling. As any model, it has limitations, which are addressed by the authors in the manuscript. Many researchers a looking forward for such model to be easily available as predictor of brain metabolite changes in various experimental setups or neurological diseases or disease models. Having this model as an easy-to-use toolbox would be outstanding.

A few points require clarification:

Blood compartment in the brain is not negligible, and blood glucose is about 5-fold larger than that in brain parenchyma. How is this brain and blood glucose concentrations handled when predicting MRS glucose ?

As I can understand, brain glycogen metabolism is not considered in the model. This certainly contributes for glucose pools when brain activity is stimulated, and it is worth discussing.

Discussion on cmro2 and glycolysis: this paragraph needs some text clarification. Since glycolysis does not metabolize glycogen, there is quite some confusion in paragraph 2 of page 15.

The second limitation presented in the discussion is not really a limitation but a model design decision.thus, it is recommended that the authors elaborate on how including vascular regulation of flow and volume would impact model’s predictions. For example, could that improve BOLD signal rise prediction better? I am referring to the discrepancy in figure 6. Or could it predict better the levels of glucose, which have a substantial contribution from the vascular compartment?

Reviewer #3: The paper uses the MEIGO software algorithm to investigate the addition of a metabolic model in an already published model of neuromuscular coupling. The authors show through the use of training data the time profiles of Lactate, Glutamate, Glucose and Aspartate following a visual stimulation of some 500 or more seconds. This stimulation is quite long and one has to wonder what other factors come into play during this time. These results it should be noted are relative changes rather than an actual baseline result.

Importantly the parameter list ( some 62 in all) as listed in Supplementary document S2 shows the "optimised" value as well as the upper and lower bounds determined during the global optimisation process.

This reviewer has to make a significant leap of faith in believing the algorithm of a global optimisation process in dimension 62 ( or above). In fact investigating the parameter list provides an argument which shows that the parameter values are either dependent on the training data or are incorrect. Looking at the parameter ky_{4}, its optimised value is 46.8257, but according to the algorithm the upper bound is 69990.4509 (still 4 decimal places ! really?) . Using equation (15) why should we believe the system is stable and of use to the physiological community. The parameter values may provide some result which matches the training data and thenceforth provide a result if other environmental changes occur but these dynamic steady states could reside anywhere in 62 dimensional space. If other parameters are perturbed by a small amount delta say will the dynamic steady state be moved by no more than a linear function of delta where that rate of change of function is small.

Secondly is the use of 10^5 samples sufficient (when using equation 17) to evaluate a data set given that the optimisation process occurs in 62 dimensions. Work from other groups suggest otherwise.

Also the equations set ( when looking at the parameter dimensions) are supposedly in dimensionless form , except time. Why was time not non-dimensionalised ?

Finally There seems to be no information on the family of models that are not rejected by the optimisation process.

**Have the authors made all data and (if applicable) computational code underlying the findings in their manuscript fully available?**

Reviewer #1: None

Reviewer #2: None

Reviewer #3: **No: **NO software available.

PLOS authors have the option to publish the peer review history of their article (what does this mean?). If published, this will include your full peer review and any attached files.

Reviewer #1: No

Reviewer #2: No

Reviewer #3: No
---

## [Decision Letter · Decision Letter 1]

30 Sep 2022

Dear Mr Sundqvist,

Thank you very much for submitting your manuscript "Mechanistic model for human brain metabolism and its connection to the neurovascular coupling." for consideration at PLOS Computational Biology.

As with all papers reviewed by the journal, your manuscript was reviewed by members of the editorial board and by several independent reviewers. In light of the reviews (below this email), we would like to invite the resubmission of a significantly-revised version that takes into account the reviewers' comments.

The reviewers have raised several problems with the revision - careful attention to their comments is recommended in order to move the manuscript along.

We cannot make any decision about publication until we have seen the revised manuscript and your response to the reviewers' comments. Your revised manuscript is also likely to be sent to reviewers for further evaluation.

Sincerely,

Anders Wallqvist

Academic Editor

PLOS Computational Biology

Daniel Beard

Section Editor

PLOS Computational Biology

The reviewers have raised several problems with the revision - careful attention to their comments is recommended in order to move the manuscript along.

Reviewer's Responses to Questions

**Comments to the Authors:**

Reviewer #1: Having a model with the "same level of complexity as the data" (authors' words) also means that the predictive power of the model is accordingly of the same level. What the physiology community cares about is the ability to make some predictions, especially about what we cannot measure.

It is no surprise that model identifiability is of no interest at all (except as an academic exercise) in modern machine learning. For instance, deep learning models have many millions of hyper-parameters, which is always much larger than the data used to train the model. But this doesn't matter, as long as the model behaves well in the test-set, because the value of the parameters is not important, as these do not have any physical meaning (so that you can have different orders, signs, etc., leading to the same results).

Biological modeling is not machine learning, and I would expect that parameter values do have a meaning.

Thus, if the authors' aim at providing a "minimal model" (their words) with clear parameter identifiability, this should be exactly the reason for having well-identified mechanistic details instead of approximate reaction rate equations (mass-action) and stoichiometries (see my previous example of the MAS), as it is in the present model.

All in all, it is not very difficult to fit the experimental time-courses of aspartate, glutamate, glucose, and lactate using a simple first-order linear model with 4 equations and 16 parameters. Unfortunately, such a minimal model would be useless, regardless of however narrow would be the parameter distributions.

Anyway, maybe the authors' aim is to address model identifiability. If so, as a reviewer I would suggest that the authors' revise the manuscript in order to put model identifiability as the main purpose of the study.

1. First, the authors should distinguish between structural and practical model identifiability.

2. Then they should provide calculations of structural model identifiability for linear (or pseudo-linear), like the mass-action used in present model, and non-linear kinetics (e.g., Michaelis-Menten). To my knowledge, this has only been demonstrated using linearized models. Furthermore, it is commonly assumed that the system has a unique (as well as locally and asymptotically stable) steady state, but this is not necessarily true in biological networks.

3. Then they should provide evidence that previously published models are unidentifiable, in order to justify the development of the present model.

4. Finally, they should demonstrate that the present model is identifiable and, most importantly, gives some clear advantage in terms of explainability of neurophysiological and biochemical processes compared to previously published models.

In summary, the authors' answer that they gave up on mechanistic knowledge because otherwise the model might be unidentifiable, cannot be accepted without a clear demonstration of such claim.

Finally, let me give a brief example of what I mean. The sole addition of glycogen metabolism (in astrocytes) by DiNuzzo et al (2010b, PMID: 20827264) to their previously published model of brain energy metabolism (DiNuzzo et al, 2010a, PMID: 19888285) provided a biochemical explanation for why astrocytes reduce glucose uptake during functional activation (for neuronal use). This mechanistic explanation (brought about by an increased complexity of the model) formed the basis for the glucose sparing by glycogenolysis (GSG) mechanism, which has been recently shown to be able to excellently fit human (fluxomics) data from more than two decades of 13C-NMR experiments (Rothman et al, 2022, PMID: 34994222), among other neurophysiological explanations. The authors' argument that "there is a limit to how much published data is relevant to consider" (authors' words) is of course dependent on what's the question underlying the developed model.

Neuron-astrocyte interactions are extremely important, as these compartments have distinct roles in neurotransmission, glutamate-glutamine cycle, and ionic movements, all aspects that are critical to brain function. Thus, it goes without saying that I am not entirely comfortable with the authors' answer that "This example shows that adding additional complexity to the model in the form of multiple compartments i) does not improve the model’s ability to explain the data we have considered in the manuscript, but ii) that it does reduce the parameter identifiability. The same problem would likely be the case if we added e.g., glycogen metabolism, or expanded the metabolic pathways that are currently lumped together into single reaction." (authors' words). Okay, adding complexity sometimes does not improve fit of the data or might even reduce parameter identifiability, so what? The physiology community is interested in neurophysiological insights, not mathematical modeling.

I understand that the authors insist on building a minimal model, and hence they take out everything that is not necessary to explain data. To me, this approach does not make any sense. I might well be wrong, and since my opinion could be now biased, it is in the interest of the authors and the Journal that I withdraw from reviewing this manuscript. I hope some of my comments were useful to improve the paper, and I apologize with the authors and with the Editor.

Reviewer #2: The authors have extended the manuscript to address my previous concerns.

Reviewer #3: In terms of the length of time for the training data the authors have unfortunately missed the point of the question. That is what happens to the parameter values if the further data becomes available but with a shorter ( or longer) time ? Would the parameters change and if so why, can the model explain this? by taking the minimal modelling approach the only outcome of the model is a set of parameters that exist for a single data set.

For the issue of the large dimensionality of the system, the previous work referenced, does this have exactly the same data set for which parameter values ( 62-18) are evaluated? If not what proof can be shown that allows a dimensional reduction of this type?

Given, as the authors admit that the parameter values can have a vast range ( "an order of magnitude") what confidence do we have in the model? Even if we take into account the values relative to baseline.

The authors say

"The interesting aspect of the parameters are therefore

captured by the totality of the many such acceptable parameter vectors. Each one of them

means close to nothing, but what they say jointly, for e.g. a prediction, is a robust and welldetermined

prediction by the model (well-determined in the sense that the prediction has

to lie within the plotted uncertainty range, in order for the model to be consistent with

available data)."

What does this mean ?

In terms of the use of 10^5 samples and the authors graph it must be noted that since the ordinate is a log scale variations that look small between 10^5 and 10^6 may not be small at all. Indeed some of the parameter variations are substantial.

**Have the authors made all data and (if applicable) computational code underlying the findings in their manuscript fully available?**

Reviewer #1: Yes

Reviewer #2: Yes

Reviewer #3: Yes

PLOS authors have the option to publish the peer review history of their article (what does this mean?). If published, this will include your full peer review and any attached files.

Reviewer #1: No

Reviewer #2: No

Reviewer #3: No
---

## [Editor Report · Decision Letter 2]

7 Dec 2022

Dear Mr Sundqvist,

We are pleased to inform you that your manuscript 'Mechanistic model for human brain metabolism and its connection to the neurovascular coupling.' has been provisionally accepted for publication in PLOS Computational Biology.

Best regards,

Anders Wallqvist

Academic Editor

PLOS Computational Biology

Daniel Beard

Section Editor

PLOS Computational Biology

---

## [Editor Report · Acceptance letter]

15 Dec 2022

PCOMPBIOL-D-22-00806R2 

Mechanistic model for human brain metabolism and its connection to the neurovascular coupling.

Dear Dr Sundqvist,

I am pleased to inform you that your manuscript has been formally accepted for publication in PLOS Computational Biology. Your manuscript is now with our production department and you will be notified of the publication date in due course.

With kind regards,

Zsofi Zombor
